# Cryptic pocket formation underlies allosteric modulator selectivity at muscarinic GPCRs

Scott A. Hollingsworth [1,4,6], Brendan Kelly [1,6], Celine Valant[2,6], Jordan Arthur Michaelis[2],
Olivia Mastromihalis[2], Geoff Thompson[2], A.J. Venkatakrishnan[1], Samuel Hertig[1], Peter J. Scammells[2],
Patrick M. Sexton [2], Christian C. Felder[3,5], Arthur Christopoulos[2] & Ron O. Dror [1]

Allosteric modulators are highly desirable as drugs, particularly for G-protein-coupled receptor (GPCR) targets, because allosteric drugs can achieve selectivity between closely related receptors. The mechanisms by which allosteric modulators achieve selectivity remain elusive, however, particularly given recent structures that reveal similar allosteric binding sites across receptors. Here we show that positive allosteric modulators (PAMs) of the M1 muscarinic acetylcholine receptor (mAChR) achieve exquisite selectivity by occupying a dynamic pocket absent in existing crystal structures. This cryptic pocket forms far more frequently in molecular dynamics simulations of the M1 mAChR than in those of other mAChRs. These observations reconcile mutagenesis data that previously appeared contradictory. Further mutagenesis experiments validate our prediction that preventing cryptic pocket opening decreases the affinity of M1-selective PAMs. Our findings suggest opportunities for the design of subtype-specific drugs exploiting cryptic pockets that open in certain receptors but not in other receptors with nearly identical static structures.

[1] Departments of Computer Science, Molecular and Cellular Physiology, and Structural Biology, and Institute for Computational and Mathematical Engineering, Stanford University, Stanford, CA 94305, USA. [2] Drug Discovery Biology, Monash Institute of Pharmaceutical Sciences and Department of Pharmacology, Monash University, Parkville, VIC 3052, Australia. [3] Eli Lilly and Co., Neuroscience, Lilly Corporate Center, Indianapolis, IN 46285, USA. [4] Present address: Merck & Co., Boston, MA 02110, USA. [5] Present address: Karuna Pharmaceuticals, Inc., South San Francisco, CA 94080, USA. [6] These authors contributed equally: Scott A. Hollingsworth, Brendan Kelly, Celine Valant. Correspondence and requests for materials should be addressed to B.K. (email: kellybj86@gmail.com) or to A.C. (email: arthur.christopoulos@monash.edu) or to R.O.D. (email: ron.dror@stanford.edu)

A major challenge in drug discovery is finding ligands that bind selectively to the desired target receptor but not to closely related off-target receptors, which are responsible for the side effects of many drugs. This challenge is particularly acute for G-protein-coupled receptors (GPCRs), the targets of approximately one-third of all current medicines[1–3]. Most GPCR drugs would ideally be designed to selectively target one of several highly similar receptor subtypes that bind the same endogenous (native) ligand. Allosteric modulators—i.e., ligands that bind at sites spatially distinct from the orthosteric site where endogenous ligands bind—have long been pursued as a means to achieve subtype selectivity[3–8], because genomic sequence analysis indicates that allosteric sites can vary more than the highly conserved orthosteric site[9]. Indeed, a number of highly selective allosteric modulators of GPCRs have been discovered. Surprisingly, however, recently solved GPCR structures indicate that allosteric sites are often structurally similar across receptor subtypes or even less closely related receptors[8,10–13], posing a conundrum as to how certain allosteric modulators actually achieve their subtype selectivity.

An excellent exemplar of this phenomenon is the muscarinic acetylcholine receptor (mAChR) family, a group of five closely related class A GPCRs (subtypes M1–M5) that have long served as models for studying allosteric modulation of GPCRs by drug-like molecules[14]. Individual mAChR subtypes are pursued as drug targets for a wide range of conditions, including schizophrenia, Parkinson's, Alzheimer's, bladder dysfunction, and chronic obstructive pulmonary disease[15–20]. The orthosteric site of mAChRs is virtually identical across subtypes, and, as a consequence, selective orthosteric ligands have proven difficult to discover and develop[10,21]. The mAChRs also have at least one allosteric site, located in an extracellular vestibule (ECV) about 15 Å above the orthosteric site (Fig. 1a). Numerous allosteric modulators that bind at this ECV site have been identified, of which several are highly selective for one subtype over

others[10,18,22–30]. However, recent crystal structures of the M1–M4 mAChRs show that this allosteric site is nearly identical in shape across these four subtypes (Fig. 1b)[10–13].

Several studies have suggested that differences in conformational dynamics within other protein families—and particularly the formation of cryptic pockets not observed crystallographically in the absence of a particular ligand—may contribute to subtype selectivity of orthosteric ligands[31–34]. The promise of identifying druggable sites that are absent in experimentally observed structures, thus creating opportunities for therapeutic design and development, has led to significant work in identifying and predicting such sites using computational methods[35–38]. Several studies have even suggested that differences in conformational dynamics within protein families may contribute to subtype selectivity of orthosteric ligands[31–34]. Very little is known, however, about whether this mechanism is operative at the largest family of drug targets—GPCRs[39]—or whether it is transferable to selective allosteric modulators.

One of the most subtype-selective GPCR modulators known is BQZ12 (3-((1S,2S)-2-hydroxycyclohexyl)-6-((6-(1-methyl-1H-pyrazol-4-yl)pyridin-3-yl)methyl)benzo[h]quinazolin-4(3H)-one), a positive allosteric modulator (PAM) that is extremely selective for M1 over M2–M5 mAChRs[27,40] (Supplementary Fig. 1). Surprisingly, the M1 mAChR crystal structure does not appear to allow for a bound pose of BQZ12 that matches existing mutagenesis data[27,28]. Even less clear is how BQZ12 and related M1-targeting allosteric modulators[22,24,26–28,30,41] achieve their subtype selectivity. We hypothesize that dynamics might play a hitherto unappreciated role in allosteric pocket formation and allosteric modulator selectivity at GPCRs.

With the aim of revealing how BQZ12 and related allosteric modulators bind and achieve subtype selectivity, we perform extensive atomic-level simulations of the M1, M2, M3, and M4 mAChRs. We identify a cryptic pocket in the M1 mAChR allosteric site that is not observed in the crystal structure but that

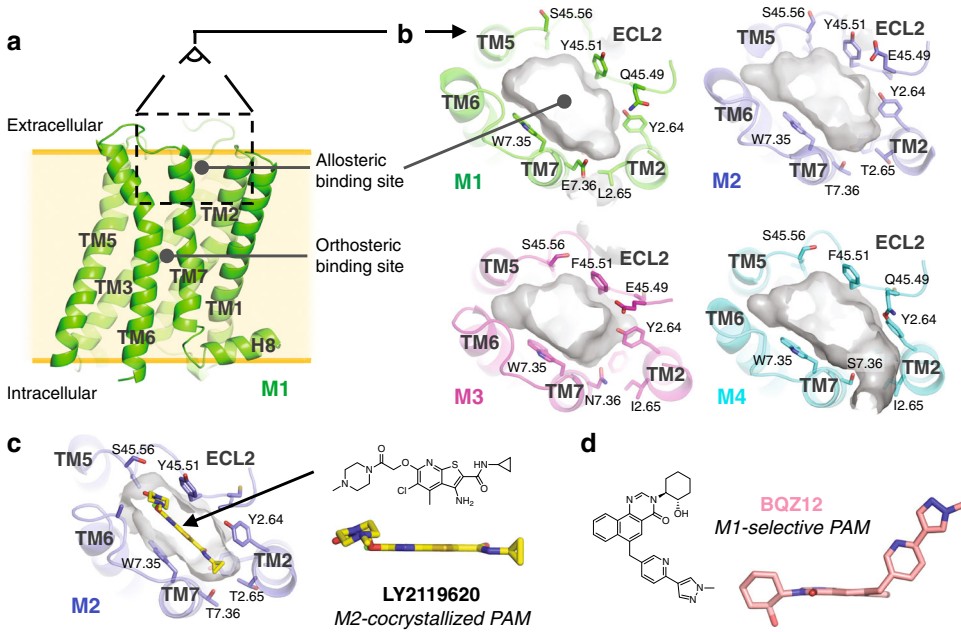

**Fig. 1** Allosteric pockets in mAChR crystal structures are similar in shape. **a** An allosteric binding site of the M1 mAChR is located on the extracellular side of the receptor, in the extracellular vestibule (ECV). This ECV allosteric site, located between ECL2, TM2, TM6, and TM7, is adjacent to but distinct from the orthosteric binding site. **b** The ECV allosteric pockets observed in the crystal structures of the inactive M1, M2, M3, and M4 mAChRs are similar in shape (pocket surface displayed in gray with selected side chains shown as sticks). **c** LY2119620, a PAM crystallized in complex with the agonist-bound active-state M2 mAChR, is planar in shape, similar to many other known mAChR PAMs. **d** In contrast to LY2119620, BQZ12 and similar M1-selective PAMs are distinctly nonplanar

opens dynamically in simulation. Identification of this cryptic pocket not only allows for determination of a BQZ12 binding pose consistent with prior mutagenesis data but also explains how BQZ12 and related modulators achieve their selectivity for the M1 mAChR. To validate our computational results, we use them to design mutations predicted to disrupt the opening of this cryptic pocket. We then confirm experimentally that these mutations reduce the binding affinity of M1-selective modulators but not the affinity of a non-M1-selective modulator. Our results demonstrate that an allosteric GPCR modulator can achieve specificity by exploiting differences in the conformational ensembles of receptors and thus suggest opportunities for the design of subtype-selective allosteric drugs even when allosteric sites appear to be similar across subtypes.

## Results

**Identification of a cryptic pocket in the M1 mAChR.** To date, crystal structures have been solved for the M1–M4 mAChRs in their inactive states bound to an orthosteric antagonist, with no allosteric modulator bound[12,13,21]. Structures have also been solved for an active-state M2 mAChR bound to an orthosteric agonist, with and without the PAM LY2119620 bound[10] (Fig. 1c). Despite these available mAChR structures, identification of a binding pose for the highly selective M1 mAChR PAM, BQZ12 (Fig. 1d, Supplementary Fig. 1), has proven challenging. In the M2 mAChR, the co-crystallized PAM LY2119620 binds in the ECV allosteric site—more specifically, in a planar pocket surrounded by transmembrane (TM) helices 2, 6, and 7 and extracellular loop 2 (ECL2)[10]. Aromatic residues W7.35 and Y45.51 (ECL2) form pi-stacking interactions with the aromatic core of LY2119620. (We use Ballesteros–Weinstein residue numbering[42,43], with 7 indicating TM 7 and 45 indicating the loop connecting TMs 4 and 5, namely ECL2.) At the M1 mAChR, mutation of either W7.35 or Y45.51 (ECL2) ablates binding for BQZ12 and its derivative BQCA (benzyl quinolone carboxylic acid), suggesting that the large aromatic cores of these ligands also stack between W7.35 and Y45.51[27,28]. However, while LY2119620 and most other known PAMs of the native mAChR agonist, acetylcholine, are planar molecules, BQZ12 is distinctly nonplanar, with a long arm extending out of the plane of the aromatic core (Fig. 1c, d). The fact that mutation of Y2.61 or Y2.64 significantly lowers the binding affinity of BQZ12[27] suggests that the nonplanar arm extends towards these TM2 residues, but the geometry of the allosteric site in the M1 mAChR crystal structure is such that there is no room to position the nonplanar arm in this direction if the aromatic core is indeed between W7.35 and Y45.51. The allosteric site in an active-state model of the M1 mAChR, while smaller in shape owing to the contraction of the ECV upon activation, is similar in that one cannot place BQZ12 in a pose that agrees with previously published experimental data (see Methods).

To probe the flexibility of the M1 mAChR allosteric site, we carried out microsecond-timescale MD simulations of the receptor, with no allosteric modulator present. We performed simulations under four conditions: the inactive state in the presence and in the absence of the co-crystallized orthosteric antagonist tiotropium (Tio), and the active state in the presence and in the absence of the endogenous orthosteric agonist acetylcholine (ACh). Analysis of the resulting trajectories revealed that the allosteric site can dynamically undergo a conformational change that significantly alters its shape (Fig. 2). Y2.64, which forms a hydrogen bond with the backbone of C178 (C45.50) in all available mAChR crystal structures, occasionally rotates away from ECL2 to form a hydrogen bond with E7.36 in simulation. E7.36, in turn, rotates inwards towards the allosteric site from its

original conformation and is then further stabilized by an additional hydrogen bond with Y2.61, located one helical turn below Y2.64 (Supplementary Fig. 4). The crystal structure of the M1 mAChR shows a salt bridge between E7.36 and K1.27, but this salt bridge breaks frequently in simulation and does not prevent rotation of E7.36 (Supplementary Fig. 5). The rotation of Y2.64 is coupled to the movement of the extracellular terminus of TM2 away from ECL2, weakening the initial hydrogen bond between Y2.64 and ECL2 residue C45.50. Together, these motions result in the opening of a secondary binding pocket adjacent to the allosteric site between TM2 and ECL2, which we refer to as the cryptic pocket. The cryptic pocket was observed to open and collapse spontaneously in simulations of the M1 mAChR under all four conditions—that is, the active state with and without ACh bound, and the inactive state with and without Tio bound (Fig. 3, Supplementary Figs. 3 and 6). While it is possible that additional factors, such as relative helix positioning or the conformation of ECL2, play a role in the formation of the cryptic pocket, the motions of Y2.64 and E7.36 appear essential in governing the stabilization of the open pocket conformation.

Unlike the crystal structure of the M1 mAChR, this MD-identified alternative conformation allows for the determination of a binding pose for a nonplanar M1-selective PAM. In the presence of the open cryptic pocket, one can position the planar aromatic core of BQZ12 or BQCA between W7.35 and Y45.51 while placing the nonplanar arm into the newly opened cryptic pocket. The resulting pose for BQZ12 agrees with previously irreconcilable mutagenesis data[27,28,44,45], and BQZ12 remained stable in this pose in multiple independent microsecond-timescale simulations (Supplementary Fig. 7).

**The cryptic pocket is open most often in the M1 mAChR.** We next set out to address how BQZ12 and BQCA achieve subtype selectivity for the M1 mAChR. To test the hypothesis that opening of the cryptic pocket may be uniquely favored in the M1 mAChR over other subtypes, we performed microsecond-timescale MD simulations of the M2, M3, and M4 mAChRs; we did not simulate the M5 mAChR because no experimental structure of that receptor is available. As with the M1 mAChR, we performed simulations of each of the subtypes under four conditions: the active state with and without ACh bound, and the inactive state with and without Tio bound. In 12 independent simulations carried out at each mAChR subtype (three simulations under each of four conditions) the cryptic pocket was open 50.2% of the time in the M1 mAChR, but only 0.4% in M2, 3.4% in M3, and 4.3% in M4 mAChRs. By contrast, the M1 mAChR cryptic pocket opening was fairly similar in the active and inactive states and in the presence and absence of ACh, suggesting that the cryptic pocket is more important for selective binding than for a PAM's ability to favor ACh binding or stabilize the active state.

The identity of the residue at position 7.36 appears to contribute to these differences in frequency of cryptic pocket formation. The residue at this position differs in each of the five mAChR subtypes (Glu in M1, Thr in M2, Asn in M3, Ser in M4, and His in M5; Supplementary Fig. 8). In simulations of the M1 mAChR, E7.36 forms a hydrogen bond with Y2.64 that keeps the cryptic pocket open. A hydrogen bond between residue 7.36 and Y2.64 rarely forms in simulations of any other subtype. In the M2 and M4 mAChRs, the interaction between Y2.64 and the backbone of ECL2 residue C45.50 rarely broke, preventing Y2.64 from rotating outwards to interact with TM7 (Fig. 3, Supplementary Fig. 6). In the M3 mAChR, Y2.64 did sometimes rotate away from ECL2 to interact with TM7, but—unlike in the M1 mAChR—this rotation did not result in cryptic pocket opening because an additional salt bridge formed between K7.32

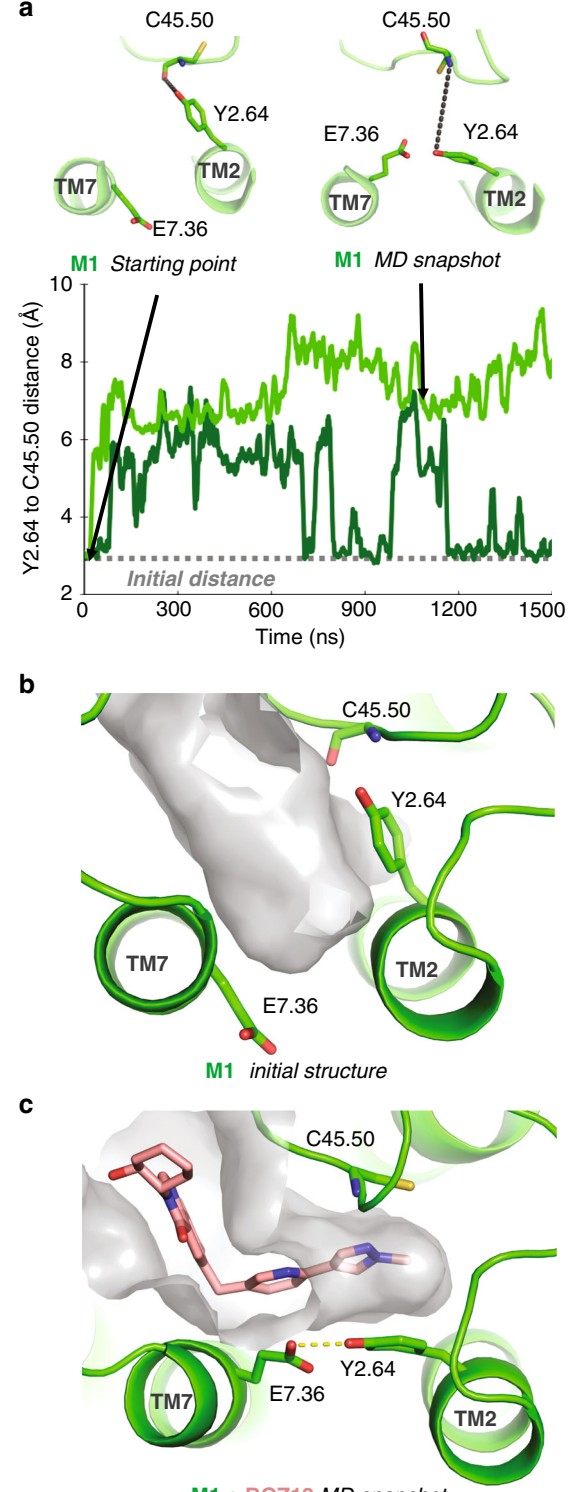

**Fig. 2** Cryptic pocket formation allows for binding of the M1-selective PAM BQZ12. **a** In simulations of the M1 mAChR, Y2.64 was observed to dynamically rotate away from the backbone of C45.50 (top, left) to interact with E7.36 (top, right). This rotation opens a cryptic pocket in the M1 mAChR allosteric binding site. The distance between Y2.64 and C45.50 from two representative simulations of M1 mAChR tracks the dynamic opening and closing of the cryptic pocket (bottom, Supplementary Fig. 2). These simulations were of an active-state receptor with neither orthosteric nor allosteric ligand bound, but inactive-state simulations and simulations with orthosteric ligands yielded similar results (Fig. 3). Snapshots shown at top are from the simulation represented by the light-green trace below. **b** The allosteric site surface in the initial active-state model of the M1 mAChR is shown in gray. **c** The opening of the cryptic pocket in simulation substantially alters the shape of the M1 mAChR allosteric site and allows for the docking of BQZ12 by placing the nonplanar arm of BQZ12 into the cryptic pocket. The resulting binding mode agrees with previous mutagenesis data (Supplementary Fig. 4)

affinity and selectivity of BQZ12 relative to BQCA[27,28]: the larger methyl-pyrazole group of BQZ12 extends further into the cryptic pocket to form a favorable pi–pi stack with Y2.64, an interaction that is only possible when the cryptic pocket is open (Supplementary Fig. 4).

**Mutagenesis confirms the importance of cryptic pocket**. In order to validate our computationally derived selectivity mechanism, we set out to reduce the ability of the cryptic pocket to open in the M1 mAChR through targeted mutagenesis. We chose to target the hydrogen bond between Y2.64 and E7.36, which helps stabilize the open cryptic pocket conformation. We eliminated the hydrogen bonding ability of Y2.64 or E7.36 via Y2.64F or E7.36A mutations, each stably expressed individually in CHO FlpIn cells (Supplementary Table 1, Supplementary Fig. 10). We also created an E7.36S mutant; the serine residue, found at this position in the M4 mAChR, has hydrogen bonding capability but would not be expected to form a stable hydrogen bond with Y2.64 due to its geometry (Supplementary Figs. 4 and 11, Supplementary Table 2). We predicted that each of these mutations would reduce the frequency of cryptic pocket opening and thus reduce the affinity of non-planar, M1-selective PAMs such as BQZ12 and BQCA. Simulations of each of these mutants resulted in a shift away from the open cryptic pocket conformation (Supplementary Fig. 12).

In radioligand binding experiments, each of these mutations led to a substantial reduction in the affinities of both BQZ12 and BQCA (Fig. 4, Supplementary Figs. 10 and 11, Supplementary Tables 1 and 2) both in the presence and in the absence of the orthosteric ligand ACh. The reductions in affinity were similar in the presence and absence of ACh; in other words, BQZ12 and BQCA maintain their PAM activity in the mutants but bind less tightly. In sharp contrast, the affinity of LY2033298—a non-M1-selective derivative of LY2119620—remained constant, or even increased slightly, in these mutants. Like LY2119620, LY2033298 is planar and would thus not be expected to bind in the cryptic pocket. This control supports the conclusion that the reduced affinity of BQZ12 and BQCA is due to disruption of cryptic pocket opening.

Taken together, these mutagenesis results validate our computational finding that the cryptic binding pocket is uniquely important in binding of non-planar, M1-selective PAMs to the M1 mAChR. We also note that, experimentally, these mutations decrease BQZ12 affinity more than BQCA affinity, in accord with our computationally derived conclusion that BQZ12 forms tighter interactions in the cryptic pocket. The fact that our simulation-

and E45.49 of ECL2, replacing the crystallographically observed Y2.64–ECL2 interaction in keeping the cryptic pocket closed (Supplementary Fig. 9).

Together, these results suggest a mechanism by which M1-selective PAMs achieve subtype selectivity. Specifically, PAMs like BQZ12 bind to a conformation of the allosteric site in which the cryptic pocket is open, and which is visited almost exclusively in the M1 mAChR subtype. This cryptic pocket-based mechanism of selectivity explains why most potent M1 mAChR PAMs are nonplanar in nature[22,24,26–28,30,41]. It also explains the increased

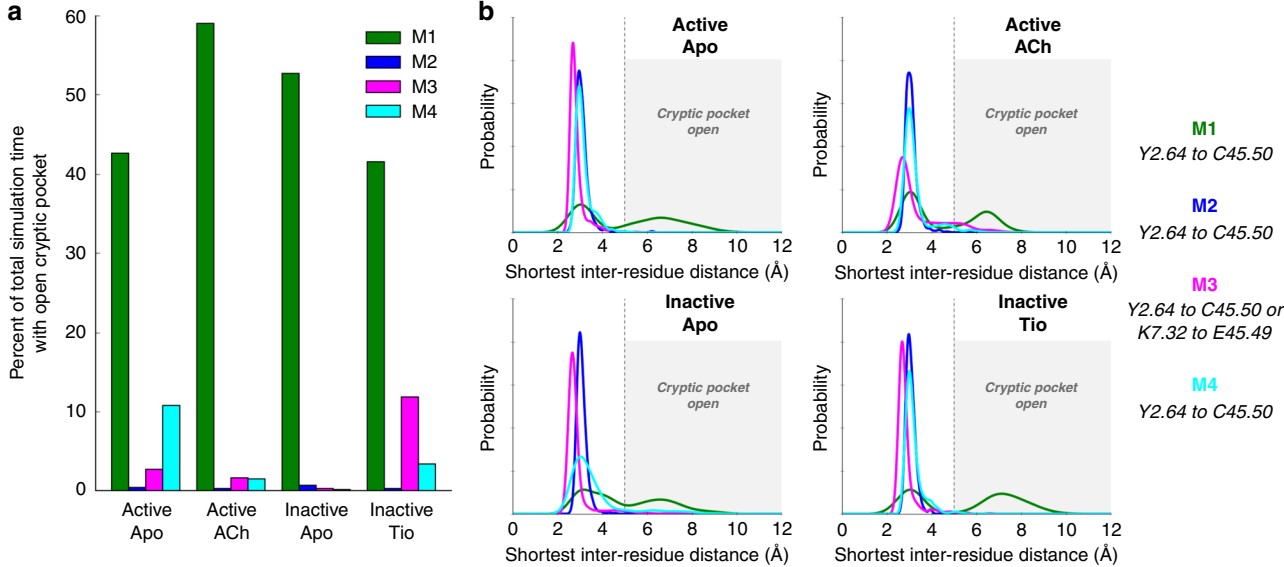

**Fig. 3** The cryptic pocket is open far more often in the M1 mAChR than in other mAChRs. **a** The cryptic pocket appears far more often in the M1 mAChR across all conditions than in M2, M3, and M4 mAChRs. Source data are provided in a Source Data file. **b** Distance histograms are displayed for each subtype across each of the four simulated conditions. Distances displayed are specified on the right. For each mAChR subtype, we carried out three independent simulations in each of four separate conditions: Active unliganded (Apo), Active acetylcholine (ACh)-bound, Inactive Apo, and Inactive tiotropium (Tio)-bound. In the M1, M2, and M4 mAChRs, we classify the cryptic pocket as open in a simulation frame if the minimum distance between non-hydrogen atoms in Y2.64 and C45.50 of ECL2 in that frame is greater than 5 Å (except for the M4 Inactive-Apo condition, where the minimum distance from Y2.64 to any ECL2 residue is used due to a register shift observed in one simulation that caused Y2.64 to interact with the backbone of Q45.49). For the M3 mAChR, we used the minimum of the Y2.64–C45.50 and K7.32–E45.49 distances, as K7.32 sometimes forms a salt bridge with E45.49 of ECL2 that also closes the cryptic pocket (Supplementary Fig. 8). As shown in Supplementary Fig. 3, tracking the opening of the cryptic pocket by monitoring the formation of the hydrogen bond between residues 2.64 and 7.36 similarly suggests that the cryptic pocket opens predominantly in the M1 mAChR

based findings predict the mutant binding results across three distinct PAMs supports a cryptic pocket-based mechanism of subtype selectivity of M1 PAMs and provides a structural explanation inaccessible from crystal structures alone.

**Ligand modification also confirms cryptic pocket importance**. To further validate the computationally derived selectivity mechanism, we rationally selected a derivative of BQZ12 to reduce the dependence of the resulting PAM on the formation of the cryptic pocket. We chose a compound (MIPS-1519; previously compound 12 in Mistry et al.[26]) lacking the terminal methyl-pyrazole group of BQZ12, thus removing the ability of the PAM to pi-stack with Y2.64 in our computationally derived binding pose while still inserting the nonplanar arm into the cryptic pocket. The computationally derived selectivity mechanism would predict that, due to the lack of pi-stacking with Y2.64, the affinity of this modified PAM would be reduced when compared to BQZ12, while binding of the PAM would remain sensitive to the cryptic-pocket-disrupting mutations due to the remaining nonplanar arm.

Indeed, MIPS-1519 was found to bind at a lower affinity than BQZ12 to the M1 wild-type mAChR both in the presence and absence of ACh (Fig. 4, Supplementary Tables 1 and 2). MIPS-1519 also showed a significant reduction in binding affinity across all tested mutants, suggesting that it still requires the nonplanar pocket to bind (Fig. 4, Supplementary Fig. 11, Supplementary Tables 1 and 2). This agrees with our computational predictions and provides additional support not only for the cryptic-pocket-based selectivity mechanism, but also for the importance of the pi-stacking interaction for the high selectivity and affinity of BQZ12 relative to other known M1 PAMs.

## Discussion

Our results highlight the role that unappreciated dynamic mechanisms—in particular, cryptic pocket formation—can play in the ability of allosteric modulators to display high selectivity in a manner that is not readily apparent in crystallographic studies or other static structures. We have focused here on mAChRs, both because they represent a popular model system for the study of GPCR allosteric modulators and because achieving subtype selectivity at mAChRs is an important step towards the development of small-molecule treatments for a wide variety of psychiatric, neurological, and peripheral disorders[40,46,47]. Although further computational and experimental work is required to determine how conformational dynamics, including cryptic pocket formation, can be leveraged to achieve selectivity at other receptors, this strategy of exploiting cryptic pocket formation to design subtype-selective allosteric modulators is likely broadly applicable both to other GPCRs and to other protein super-families. As the computational resources required for studies such as this continue to become more widely accessible[48], we expect that the strategies employed here will become more common in future therapeutic discovery and design.

## Methods

**Molecular dynamics simulation system setup**. Molecular dynamics simulations of the inactive states of the M1, M2, M3, and M4 mAChRs were initiated from the respective inactive-state crystal structures (M1, 5CXV; M2, 3UON; M3, 4U15; M4, 5DSG)[13,21,49]. For inactive-state unliganded (Apo) simulations, co-crystallized ligands were removed. In inactive-state tiotropium (Tio) bound simulations, the crystallographic coordinates of Tio were used, with the exception of the M2 mAChR, for which the co-crystallized antagonist QNB was replaced with Tio[21] (for consistency, we wished to perform all inactive-state antagonist-bound simulations with the same antagonist). The experimentally determined M2 mAChR active-state structure (4MQT) was used as the starting point for M2 mAChR active simulations, with the co-crystallized orthosteric agonist iperoxo removed for Apo

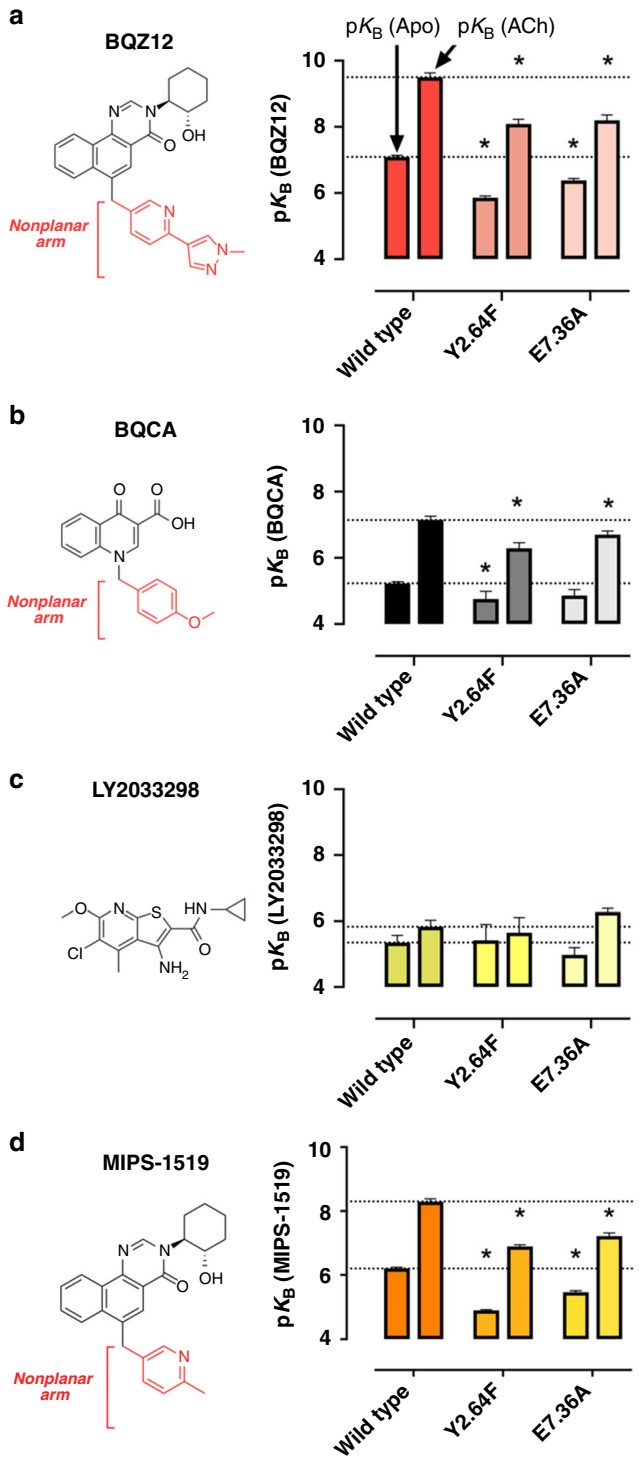

**Fig. 4** Mutational disruption of the cryptic pocket reduces affinity for M1-selective PAMs but not for a non-M1-selective PAM. **a** Experimentally determined affinities for the M1-selective PAM BQZ12 at the M1 wild-type mAChR and two mutants, Y2.64F and E7.36A. The left bar in each pair is the affinity in the absence of an orthosteric ligand (denoted $pK_B$(Apo)), and the right bar is the affinity for an ACh-occupied receptor (denoted $pK_B$(ACh)). **b** Corresponding affinities for the M1-selective PAM BQCA. **c** Corresponding affinities for the non-M1-selective PAM, LY2033298. **d** Corresponding affinities for MIPS-1519, which lacks the nonplanar methyl-pyrazole of BQZ12. Affinity estimates and their standard errors (shown by error bars) were derived by a global least-squares fit of an allosteric ternary complex model to 3 independent experiments for all conditions except BQZ12 Y2.64F, BQZ12 E7.36, LY2033298 Y2.64F, and MIPS-1519 Y2.64F which represent 4 independent experiments (Supplementary Figs. 1 and 10), with the constraint that each parameter be shared between the experiments to yield a single best estimate of the parameter and its associated standard error, as derived from the nonlinear regression algorithm. This global pooled analysis approach ensured model convergence in all instances. *, $pK_B$(Apo) or $pK_B$(ACh) significantly different ($p < 0.05$) from wild type as determined by one-way ANOVA with Dunnett's post hoc test. Source data are provided in a Source Data file

observed in modulator-free simulations of the M1 mAChR, and BQZ12 subsequently docked into the allosteric site (prior to the start of simulation).

For each structure, we first removed any additional crystalized proteins (such as nanobodies or lysozyme) and all other non-receptor molecules with the exception of select ligands described above, and retained crystallographic waters. Prime (Schrödinger) was used to model missing side chains and loops (with the exception of ICL3, which was omitted from each receptor), and neutral acetyl and methylamide groups were added to cap protein termini. We retained titratable residues in their dominant protonation state at pH 7.0, except for residues D2.50 and D3.49, whose protonation state may change upon receptor activation[50,51]. These two residues were set to different protonation states in different simulations (Supplementary Table 3). In particular, we ran simulations of the active-state M1 mAChR with both D2.50 and D3.49 protonated and unprotonated, both with and without ACh bound; the protonation state of these residues does not appear to affect cryptic pocket opening in simulation. Histidines were represented with hydrogen on the epsilon nitrogen (following manual inspection to ensure that addition of hydrogen to the delta nitrogen instead would not help to optimize the local hydrogen bond network).

The prepared protein structures were aligned on the transmembrane helices to the Orientation of Proteins in Membranes (OPM) database[52] structure of PDB 3UON. The aligned structures were then inserted into a pre-equilibrated palmitoyl-oleoyl-phosphatidylcholine (POPC) bilayer using Dabble[53]. Sodium and chloride ions were added to neutralize each system for a final concentration of 150 mM. Water box dimensions for each system were chosen to maintain at least an 18 Å buffer between protein images in the $z$ direction, while bilayer dimensions were chosen to maintain at least a 35 Å buffer between proteins in the $x$–$y$ plane of the membrane. The final systems, which varied in number of atoms and size, are listed in Supplementary Table 3.

**Simulation protocols.** For all simulations, we used the CHARMM36 parameter set for proteins, lipids, and salt ions, and the CHARMM TIP3P model for water[54–58]. Parameters for ACh were generated using the CHARMM General Force Field with the ParamChem server[59–62]. Parameters for Tio were based on previously published parameters for $N$-methylscopolamine[11], with additional parameters supplied by ParamChem. Ligand parameters are available upon request.

Three independent simulations were performed for each condition listed in Supplementary Table 3, for a total of over 157 ms of simulation across all subtypes and conditions. For all active-state simulations, each prepared structure was overlaid with the experimentally determined β2-Gs complex (PDB 3SN6)[63]. All mAChR residues that were found to be within 5 Å of Gs in the β2-Gs complex following the overlay had a 5 kcal mol$^{-1}$ Å$^{-2}$ harmonic restraint to the initial position placed on the respective residue heavy atoms to ensure the receptor would remain in the active state throughout simulation. No restraints were used in inactive-state simulations, as these remain in the inactive state unrestrained.

Simulations were performed using the CUDA-enabled version of PMEMD in Amber16 on one to two graphical processing units (GPUs)[64]. Each system underwent a similar equilibration and minimization procedure. Systems were heated in the NVT ensemble from 0 to 100 K over 12.5 ps, then from 100 to 310 K over 125 ps with 10 kcal mol$^{-1}$ Å$^{-2}$ harmonic restraints on all non-hydrogen lipid and protein atoms. Systems were then equilibrated in the NPT ensemble at 1 bar,

simulations and replaced by ACh for ACh-bound simulations[10]. Homology models of the active states of the M1, M3 and M4 mAChRs were constructed using Prime (Schrödinger) based on the available M2 mAChR active-state structure (4MQT) for the transmembrane region and the M1, M3, and M4 mAChR inactive-state structures for all extracellular and intracellular loops. Overlays of the inactive- and active-state models used as starting points for simulation are displayed in Supplementary Fig. 13. ACh was modeled into the orthosteric site for all active-state ACh-bound conditions by first docking using Glide (Schrödinger) and then selecting a pose that replicated key ligand–receptor interactions observed in mAChR crystal structures, namely the salt bridge between the cationic choline group and D3.32 and hydrogen bonding between the acetyl group and the N6.52 side chain. Simulations of BQZ12 bound to the M1 mAChR were initiated from the active-state model of the M1 mAChR described above, but with Y2.64 and E7.36 rotated to form an interhelical hydrogen bond to open the cryptic pocket as

with a 5 kcal mol$^{-1}$ Å$^{-2}$ harmonic restraint placed initially on all heavy protein atoms and reduced in a stepwise fashion by 1 kcal mol$^{-1}$ Å$^{-2}$ every 2 ns for a total of 10 ns and then by 0.1 kcal mol$^{-1}$ Å$^{-2}$ every 2 ns for an additional 20 ns. Production simulations were carried out in the NPT ensemble at 310 K and 1 bar using a Langevin thermostat for temperature coupling and a Monte Carlo barostat for pressure coupling. The majority of simulations employed a timestep of 2.5 fs, while others employed a time step of 4 fs with hydrogen mass repartitioning[65] (see Supplementary Table 3 for details). All bond lengths involving hydrogen atoms were constrained by SHAKE. Non-bonded interactions were cut off at 9.0 Å, while long-range electrostatic interactions were calculated using the particle mesh Ewald (PME) method with an Ewald coefficient of approximately 0.31 Å and an interpolation order of 4. The Fast Fourier Transfer (FFT) grid size was chosen such that the width of a single grid cell was approximately 1 Å. Trajectory snapshots were saved every 200 ps.

**Analysis protocols for molecular dynamics simulations.** The AmberTools15 CPPTRAJ package was used to reimage and center all resulting trajectories[66]. Simulations were visualized and analyzed using Visual Molecular Dynamics (VMD)[67]. Time traces from individual simulations, such as those displayed in Fig. 2, were smoothed using a moving average with a window size of 50 ns and visualized using xmgrace or the PyPlot package from Matplotlib.

To track the conformational state of the cryptic pocket, several different metrics were initially considered at the M1 mAChR subtype. In the absence of a PAM in simulation, W7.35 and Y45.51 are free to rotate, which can significantly alter the shape of the allosteric site, complicating the development of a single metric to track the overall shape of the pocket in simulation. Following extensive exploration of quantification metrics for cryptic pocket opening, two distance metrics were found to correlate with the opening of the cryptic pocket: the distance between Y2.64 and the backbone of C45.50 (the initial hydrogen bond present in the starting structures that must be broken to open the pocket) and the potential hydrogen bond distance between Y2.64 and E7.36 (the hydrogen bond that stabilizes the open pocket conformation). When applying either of these distance metrics to other mAChR subtypes, further adaptions had to be made due to sequence differences and a rare event captured in an M4 mAChR simulation. At the M2 mAChR, both distance metrics were used without any additional development. At the M3 mAChR, despite the formation of a hydrogen bond between Y2.64 and N7.36, an M3-specific salt bridge between K7.32 and E45.49 still prevented the pocket from opening to the allosteric site. For this reason, the Y2.64–C45.50 distance metric was adapted to measure the shortest distance between either Y2.64 and C45.50 or K7.32 and E45.49 (Fig. 3, Supplementary Fig. 9). In the M4 mAChR Active-ACh, Active-Apo, and Inactive-Tio conditions, the M1-derived metrics were successfully applied to monitor the cryptic pocket. However, in one M4 mAChR Inactive-Apo simulation a register shift in ECL2 allowed for Y2.64 to interact with the backbone of Q45.49 while keeping the pocket closed. For this reason, the shortest distance between Y2.64 and ECL2 (45.49–45.51) was used as a metric.

**Generation of stable cell lines and culture conditions.** Chinese Hamster Ovary (CHO-FlpIn) cells were stably transfected with the receptor constructs of interest. All cell lines were maintained in DMEM supplemented with 5% FBS, and allowed to grow at 37 °C in a humidified incubator with 5% CO$_2$. Cells were then split at a 1:10 ratio every 3 days for either assaying or maintenance of the lines. All cell lines were generated in-house in a Chinese Hamster Ovary background using the FlpIn insertion method[11], using CHO-FlpIn cell lines purchased from Invitrogen/ Thermo Fisher (catalog number R75807)

**Whole cell [³H]NMS radioligand binding assays.** Radioligand binding experiments were performed on CHO-FlpIn whole cells stably expressing the human mAChR constructs as indicated in the main text. After plating 20,000 cells in complete DMEM into 96-well ISOPLATE TC plates, cells were allowed to grow overnight at 37 °C. The next day, cells were washed twice with phosphate-buffered saline (100 mL) and re-suspended in binding buffer (10 mM HEPES, 100 mM NaCl, 10 mM MgCl$_2$, pH 7.4). Assay mixtures, in a total volume of 100 μL with a 1/10 dilution of drug, were incubated at room temperature (22 °C) for 16 h. Assays were terminated by buffer removal followed by rapid washing, twice, with ice-cold 0.9% NaCl (100 μL). Plates were allowed to dry inverted for 10 min; OptiPhase Supermix scintillation cocktail (100 μL) was added, plates were sealed (TopSeal), and radioactivity was measured in a MicroBeta2 LumiJET microplate counter. Saturation binding experiments were performed in the absence or presence of atropine (10 μM) with 0.003–3 nM [³H]NMS to assess radioligand affinity ($K_A$). Inhibition binding experiments were performed with $K_A$ concentrations of the radioligand, in most instance 0.2 nM [³H]NMS, in the presence of various concentrations of the endogenous ligand, ACh, and increasing concentrations of each mAChR PAM.

**Data analysis.** Experimentally determined datapoints from the radioligand binding assays are expressed as means ± standard errors (Supplementary Figs. 1a–e, 10, and 11 (left-hand side)).

To determine affinity and cooperativity values, computerized nonlinear regression was performed using the program Prism 7.0b (GraphPad Software, San Diego, CA). All affinities and cooperativity parameters were estimated as logarithms. Competition binding curves between [³H]NMS and ACh in the absence or presence of increasing concentrations of PAMs, were fitted to the following version of an allosteric ternary complex model:

$$Y = \frac{100 \times [A]}{[A] + \left(\frac{K_A \cdot K_B}{\alpha'[B] + K_B}\right)\left(1 + \frac{[I]}{K_I} + \frac{[B]}{K_B} + \frac{\alpha[I][B]}{K_I \cdot K_B}\right)} \quad (1)$$

where $Y$ is percentage (relative to vehicle control) specific binding, $[A]$, $[B]$, and $[I]$ are the concentrations of [³H]NMS, PAM, and ACh, respectively; $K_A$ is the equilibrium dissociation constant of [³H]NMS; $K_B$ is the equilibrium dissociation constant for the PAM at the free receptor (also referred to herein as $K_B$(Apo)); $K_I$ is the equilibrium dissociation constant of Ach; and $\alpha'$ and $\alpha$ are the cooperativities between PAM and [³H]NMS or ACh, respectively, i.e., $\alpha' = K_B$(Apo)/$K_B$(NMS) and $\alpha = K_B$(Apo)/$K_B$(ACh), where $K_B$(NMS) denotes the equilibrium dissociation constant of the PAM at the NMS-occupied receptor, whereas $K_B$(ACh) denotes the equilibrium dissociation constant of the PAM at the ACh-occupied receptor. All equilibrium dissociation constant values were estimated as or (where indicated) fixed as negative logarithms.

In order to ensure model convergence in all instances, the allosteric ternary complex model was globally fitted to all individual datasets for each allosteric modulator at each receptor construct, with the parameters constrained to be shared across these datasets. The resulting reported parameter estimates thus represent the least-squares best-fit value of each parameter, with its associated standard error as reported from the nonlinear regression program, based on the pooled analysis of the individual datasets.

**Reporting summary.** Further information on research design is available in the Nature Research Reporting Summary linked to this article.

## Data availability
The source data underlying Figs. 3a and 4a–d and Supplementary Figs. 1a–f, 10a–l, and 11a–d are provided as a Source Data file. Simulation trajectories and all other data supporting the findings of this study are available from the corresponding authors upon reasonable request.

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

## Acknowledgements

The authors thank Erik Hembre and many past and present members of the Dror Lab for countless thoughtful and helpful discussions throughout this study. This work was supported by Eli Lilly and Company through the Lilly Research Award Program (to R.O.

D.), by NIH grant R01GM127359 (to R.O.D.), by NIH Institutional Biomedical Informatics Training Grant T15-LM007033-33 Postdoctoral Fellowship (to S.A.H.), by Program Grant APP1055134 of the National Health and Medical Research Council (NHMRC) of Australia (to A.C. and P.M.S.), by NHMRC Project Grant APP1082318 (to C.V.), and by Australian Research Council Future Fellowship FT140100114 (to C.V.). A. C. is a Senior Principal, and P.M.S., a Principal, Research Fellow of the NHMRC. C.V. is an Australian Research Council Future Fellow.

## Author contributions

B.K., S.A.H., C.V., C.C.F., A.C., and R.O.D. designed research. B.K., S.A.H., C.V., J.A.M., O.M., G.T., P.J.S., A.J.V., and S.H. performed research. S.A.H., B.K., C.V., A.J.V., S.H., C. C.F., P.M.S., A.C., and R.O.D. analyzed data. S.A.H., B.K., C.V., A.C., and R.O.D. wrote the manuscript.

## Additional information

**Competing interests:** The authors declare no competing interests.

