## [Peer Review File · Nature Communications]

Reviewers' comments:

Reviewer #1 (Remarks to the Author):

This is an interesting work exploring how subtype selectivity of GPCR allosteric modulators can be achieved by occupying a dynamic pocket that is absent in the crystal structures. The authors have used extensive microsecond-scale molecular dynamics simulations to identify a selective cryptic pocket in the muscarinic M1 receptor that does not exist in the other subtypes and validated the predictions using mutagenesis and radioligand binding.

This is a novel and interesting topic both as a prove-of-concept and for the importance of designing selective molecules for the muscarinic receptor subtypes. Computational methodology is robust and the experiments adequate. The article is well written, and the results elegantly presented. I thus recommend publication after addressing a few points:

- 1) The crystal structure of the M1 shows that the counterpart of E7.36 is K1.27. Breakage of these ionic interaction is thus a necessary step so that E7.36 is free to rotate. However, this residue is not even mentioned in the text. I think it is important that the interaction E7.36-K1.27 is displayed in Fig 2A and its time-evolution also shown in Fig 2B.
- 2) Fig 2B shows many events of opening/closing of the cryptic pocket in the M1. I was surprised to see (in Suppl Fig 3) that this occurs only in one of the three replicas. The other two show one or zero opening/closing events. The authors should comment on why this happens and clarify this in the text since otherwise the reader gets the impression that Fig 2B is representative of the three replicates (as it is erroneously stated in the figure caption) which is not.
- 3) In the methods section (line 325) it is written regarding the protonation states of D2.50 and D3.49 that they were set to "different protonation states in different simulations". What does this mean? Different protonation states in different states, in different subtypes or in different replicates? In the later case, could this be related with the differences observed between replicas? Similarly, the sentence starting with "Histidines were represented with hydrogen ..." does not provide much information. It would be clearer to write which specific His and in which subtypes were set to delta.

4) As shown in Fig 1A, the pocket described is highly exposed and the N-termini of the different muscarinic receptor subtypes (20-70 residues) that is not solved in the crystal structures could well be part of it as well. Have the authors considered this possibility?

Reviewer #2 (Remarks to the Author):

In this manuscript, Hollingsworth et al. suggest a mechanism by which allosteric modulators for muscarinic acetylcholine receptors can achieve selectivity for the M1 subtype. The authors performed and analyzed long-scale molecular dynamics simulations of all the muscarinic receptor subtypes in different states, and hypothesize that a 'cryptic' binding pocket –absent in the available crystal structures– can spontaneously open in the extracellular domains of the M1 subtype to accommodate a positive allosteric modulator.

According to the simulations, the opening/closing of this cryptic binding pocket in the M1 subtype is due to the labile nature of a hydrogen bond interaction between Y2.64 and C45.50. In the simulations, Y2.64 can relocate away from C45.50 (in the second extracellular loop) and towards E7.36 (in TM7) thus opening the cryptic pocket. Due to the diverse nature of position 7.36 in muscarinic acetylcholine receptor subtypes, the rearrangement of Y2.64 and opening of the cryptic binding pocket is not favored in subtypes M2-M5, thus explaining the selectivity of certain PAMs for the M1 subtype.

Both the scientific hypothesis underlying this work (allosteric modulators can recognize transient binding pockets that are not necessarily visible in crystal structures) and the experimental design to test it (long scale molecular dynamics simulations and site-directed mutagenesis) are very straightforward. Also, the number and length of of the simulations performed is impressive. However, the authors use surprisingly few data from the simulations to substantiate their claims. The nature of the cryptic binding pocket is simply described in terms of the Y2.64-C45.50 distance (in Figures 2A, 3A, and 3B, which contain all the computational data in the main text, presenting data derived from this single parameter). I am convinced that there must be more information in the trajectories that can provide further insights on the nature of the cryptic pocket; it seems a colossal effort to run over 127 microseconds of simulations to monitor a single atom-atom distance.

The site-directed mutagenesis data seem pretty straightforward, suggesting that disrupting the interaction that keeps the cryptic pocket open (with the single-point mutations Y2.64F and E7.36A)

reduces the frequency of the open state, as the affinity of these mutants for the PAMs is reduced (about 1 log unit) while retaining its activity as a PAM.

Despite the clarity of the goals and the presented data, I would expect a more thorough characterization of the structure/dynamics of the cryptic pocket in the MD trajectories, and perhaps a more complete analysis by site-directed mutagenesis. It is conceivable that are other factors involved in the formation/stabilization of the cryptic binding pocket in the M1 muscarinic receptor.

This work would also greatly benefit from an attempt (at least) to predict the existence of cryptic pockets in other receptors. If I understand properly, the authors imply that the presence of a cryptic pocket in the M1 muscarinic receptor could not have been predicted; is this so? In this case, are long-scale molecular dynamics simulations a requisite for finding such cryptic pockets? If so, their study will be restricted to just a few research groups worldwide, considerably limiting its usefulness to design subtype-specific allosteric drugs that exploit cryptic pockets. Perhaps the analysis of the MD trajectories allows a retrospective analysis on available sequences and/or crystal structures that allows to predict the presence of cryptic pockets? I understand that this may require much more computational and experimental work, but without a generalization of their results, the presented data represents a single example in a very specific protein family.

In summary, the hypothesis presented in this work is novel and interesting for researchers in the area of molecular pharmacology, particularly in the field of GPCRs. While the work has the potential to be influential in these fields, the present analysis of the computational and experimental data seems too limited to fully support the claims of the authors, in particular regarding the extension of the idea of cryptic pockets for allosteric modulators in other GPCRs.

Reviewer #3 (Remarks to the Author):

The manuscript „Cryptic pocket formation underlies subtype selectivity of GPCR allosteric modulators“ by Ron Dror and Arthur Christopoulos and colleagues reports on the identification of a ‘cryptic pocket’ in the allosteric binding site of the muscarinic M1 receptor (M1R). This pocket is not visible in the inactive crystal structure of the M1R but forms spontaneously as assessed by microsecond-timescale computational simulations. Interestingly, this pocket seems to form far more often in the M1R than in other muscarinic receptor subtypes. The authors claim that this ‘cryptic pocket’ dictates subtype selectivity of a previously characterized M1R positive allosteric modulator (PAM), i.e. BQZ12, a structural analogue of the well-characterized M1R PAM BQCA. Based on these

data the authors aim to generalize their findings and suggest that subtype selectivity of allosteric modulators can be achieved by exploiting differences in the conformational ensemble of GPCRs.

Overall, this is a very interesting manuscript. Novel strategies to design subtype selective drug candidates are needed and this manuscript offers a powerful approach by combining cutting-edge computational methods with more classical pharmacology. From a methodological point of view, the approach undertaken here will likely have a great impact on other GPCRs and related drug targets. While I find the approach very intriguing, powerful and of broad interest, I am more skeptical about some specific conclusions drawn from the data and about the generalizability of specific findings of this manuscript.

Therefore, I cannot recommend publication of this manuscript in its current state. However, I would be happy to look at a suitably revised version of this manuscript. Please find my comments below.

1) I am not convinced that cryptic pocket formation is independent of the state (apo, inactive inverse agonist-bound, active agonist-bound) of the M1R. My concern is based on two lines of evidence:

a. GPCR activation involves a process called 'allosteric coupling' where conformational changes at the intracellular part of the receptor (e.g. outward tilt of TM6) are allosterically linked to the conformation of the ligand binding pocket and extracellular domains (PMID: 27362234). In particular, the M2R undergoes major extracellular conformational changes which result in a complete closure of the ligand binding pocket including a 'contraction' of the allosteric binding site (also visible in Figure 1b vs 1c of the current manuscript). However, the authors state that the 'geometry of the allosteric site in an active-state model ... is similar' (page 7, lines 123-124) to the one seen in the inactive crystal structure. This is in contrast to all available active structures which show a (more (e.g. M2R) or less (e.g. β 2AR) pronounced) contraction of the ligand binding pocket and rearrangement of extracellular domains. How 'active' is the M1R active-state model reported here? Please describe, show and validate the active state M1R model. This would be an interesting finding and should be highlighted.

b. BQZ12 cannot be docked into the inactive M1R crystal structure (e.g. page 5; lines 86-88). However, the authors have successfully docked BQZ12 into an active state M1R model in a previous paper (PMID: 25326383). According to this, one would expect structural changes in the allosteric binding site to occur upon receptor activation (see above). This is contradictory to the authors' statement that the 'geometry of the allosteric site in an active-state model ... is similar' (page 7, lines 123-124) to the one seen in the inactive crystal structure. Please clarify.

2) On page 13, lines 261-263 the authors state 'We predicted that each of these mutations would reduce the frequency of cryptic pocket opening and thus reduce the affinity of the non-planar, M1-selective PAMs such as BQZ12 and BQCA'. Whereas the pharmacological data show indeed that mutation of Y2.64F and E7.36A reduce PAM affinity, the manuscript does not provide evidence that the frequency of cryptic pocket opening is reduced by the mutations. Additional

molecular dynamics simulations of both receptor mutants bound to ACh and BQZ12 should be run to provide data for the above-mentioned statement.

3) Figure 4: Many of the presented results have been published earlier. This is not uncommon as the same mutation can be studied under totally different perspectives in different manuscripts. However, in the case of BQCA, there are published results which contradict the findings presented here: in a previous report by the same group (PMID: 25326383), the influence of the Y2.64F mutation on BQCA affinity has been studied. It was concluded that this mutation does not influence BQCA affinity (pK_b(BQCA): 4.82 ± 0.06 and 4.66 ± 0.00 at wildtype and Y2.64F mutant receptors, respectively). In the current manuscript the authors show that this mutation does indeed significantly reduce BQCA affinity (Figure 4b. pK_b(BQCA): 5.23 ± 0.05 and 4.76 ± 0.23 at wildtype and Y2.64F mutant receptors, respectively). It seems that the affinity of BQCA to the free receptor varies across studies, e.g. 4.82 ± 0.06 (PMID: 25326383), 4.49 ± 0.09 (PMID: 24443568) and 5.23 ± 0.05 (this study). Please clarify these contradictory results.

4) On page 8, line 147-149 the authors state that 'simulations of BQZ12 in this pose remained stable in the allosteric site in multiple independent microsecond-timescale simulation'. Where are these data? I just find a list of simulations times in Supplementary Table 3. It would be very important to show these simulations and describe and assess the frequency and stability of interactions between BQZ12 and receptor residues from the cryptic pocket and residues found to interact with BQZ12 in a previous publication (PMID: 2532638).

5) The data provided in this manuscript do not convince me that cryptic pocket formation is a general mechanism which applies to many PAMs at the M1R, related muscarinic subtypes, or even other Class A GPCRs. The authors mention several times that BQZ12 'and related/similar PAMs (e.g. page 4, line 73; page 5, line 89; page 5, line 94) are nonplanar in nature (page 11, 220-221)'. However, the data shown in the manuscript are solely based on BQZ12 itself and BQCA (although there are contradictory results in the literature for BQCA, c.f. point 3). Which 'related' PAMs do the authors refer to? It would be very important to demonstrate both computationally and pharmacologically that these 'related' PAMs bind to the cryptic pocket. Otherwise, the findings reported here are a special case for BQZ12 (and maybe BQCA) at the M1R.

6) The authors state in the title that 'cryptic pocket formation underlies subtype selectivity'. However, subtype selectivity of BQCA has been shown to arise from positive cooperativity with ACh at M1R rather than high affinity. Please specify what mechanism of selectivity underlies which PAM.

7) Figure 2: I suggest to add a second, more mechanistic, surrogate parameter to describe the dynamics of 'cryptic pocket formation'. Estimation of the distance between E7.36 and Y2.64, i.e. the formation of the H-bond depicted in Figure 2C, would be helpful in this regard.

8) Figure 1: The authors state that the allosteric pockets 'are similar in shape' based on the shape of the pocket surface. How is this pocket surface measured exactly? To me the pockets do look fairly distinct, e.g. the pocket surfaces of M2R, M3R and M4R, but not M1R, extend between TM5 and TM6; and the pocket surface of M4R covers part of the space between TM2 and TM7. On what parameters is the degree of similarity based? Another report claimed that the allosteric pockets are distinct (PMID: 27490498). Please discuss.

9) The authors state that 'allosteric sites are often very similar structurally across receptor subtype' (page 3, lines 52-53). However, this statement can only be drawn from muscarinic receptor structures. The wording suggests that this would be a general feature of all GPCRs. Please rephrase.

10) The title suggests that cryptic pocket formation would be a general mechanism for all GPCR allosteric modulators. However, the data provided in this manuscript do not support this statement (see above). Based on the data, cryptic pocket formation would be, at best, a mechanism applicable to BQZ12 and BQCA. Please rephrase.

Response to Reviewers' Comments:

We thank all of the reviewers for their thoughtful feedback on our submission. We have carefully considered their comments and have made revisions that we believe significantly strengthen our manuscript. Below, we have included the reviewers' original comments (indented and in italics), followed by our responses.

Reviewer #1

This is an interesting work exploring how subtype selectivity of GPCR allosteric modulators can be achieved by occupying a dynamic pocket that is absent in the crystal structures. The authors have used extensive microsecond-scale molecular dynamics simulations to identify a selective cryptic pocket in the muscarinic M1 receptor that does not exist in the other subtypes and validated the predictions using mutagenesis and radioligand binding.

This is a novel and interesting topic both as a prove-of-concept and for the importance of designing selective molecules for the muscarinic receptor subtypes. Computational methodology is robust and the experiments adequate. The article is well written, and the results elegantly presented. I thus recommend publication after addressing a few points:

We appreciate the reviewer's favorable comments on our manuscript.

1) The crystal structure of the M1 shows that the counterpart of E7.36 is K1.27. Breakage of these ionic interaction is thus a necessary step so that E7.36 is free to rotate. However, this residue is not even mentioned in the text. I think it is important that the interaction E7.36-K1.27 is displayed in Fig 2A and its time-evolution also shown in Fig 2B.

In all inactive-state simulations, in which we have included K1.27, the interaction between K1.27 and E7.36 breaks spontaneously. K1.27 is absent in our active-state models due to the lack of structural homology for K1.27 in the active-state structure of M2 (on which all active-state models in our study are based). Because this interaction between K1.27 and E7.36 breaks frequently and appears weak in simulation, we do not believe that it plays an important role in formation of the cryptic pocket.

As all of Figure 2 currently involves snapshots and traces taken from active-state simulations, we have chosen not to include K1.27 in Figure 2 as it is absent in our active-state models. We have instead included the time evolution of the K1.27-E7.36 interaction in all of the inactive-state M1 mAChR simulations in a new supplementary figure (Supplementary Fig. 5). We have also added a new sentence on page 8 to address this finding in the main text.

2) Fig 2B shows many events of opening/closing of the cryptic pocket in the M1. I was surprised to see (in Suppl Fig 3) that this occurs only in one of the three replicas. The other two show one or zero opening/closing events. The authors should comment on why this happens and clarify this in the text since otherwise the reader gets the impression that Fig 2B is representative of the three replicates (as it is erroneously stated in the figure caption) which is not.

While we originally chose the highlighted trace to demonstrate the dynamic nature of the Y2.64-E7.36 interaction, we agree with the reviewer that it is not the best representative of what we typically observe in simulation across all M1 mAChR conditions. We have changed the trace that we highlight in Figure 2A to a simulation that opens once to better reflect what we observe more often in simulation.

3) In the methods section (line 325) it is written regarding the protonation states of D2.50 and D3.49 that they were set to “different protonation states in different simulations”. What does this mean? Different protonation states in different states, in different subtypes of in different replicates? In the later case, could this be related with the differences observed between replicas? Similarly, the sentence starting with “Histidines were represented with hydrogen ...” does not provide much information. It would be clearer to write which specific His and in which subtypes were set to delta.

To test for potential effects of the D3.49 and D2.50 protonation states on the cryptic pocket, we ran simulations of the active-state M1 mAChR with both of these residues protonated and with both unprotonated (both with and without ACh bound). We have clarified this point in the revised Methods section. Analysis revealed that the protonation states of these residues had little effect on the shape or dynamics of the cryptic pocket.

All histidines were protonated on the epsilon nitrogen rather than delta. For each of the systems built for simulation, we analyzed the hydrogen bond network to determine whether epsilon protonation would be more likely, but this was never the case. We have revised the Methods section to clarify this point.

4) As shown in Fig 1A, the pocket described is highly exposed and the N-termini of the different muscarinic receptor subtypes (20-70 residues) that is not solved in the crystal structures could well be part of it as well. Have the authors considered this possibility?

We have indeed considered the possibility that the N-terminus may play a role. However, we would expect that the N-terminus would be highly dynamic and not make any significant, long-lasting interactions with the core of the receptor. In addition, as shown below, previous experimental work performed at the M2 mAChR has shown that removal of the receptor's N-terminus (“M2 N-ter truncated”) had minimal effect on a PAM, LY2033298, which is known to bind in the extracellular vestibule. Specifically, we saw no change in the allosteric modulator's affinity (pK_B) or cooperativity ($\alpha\beta$) with the agonist, oxotremorine M (Oxo-M), relative to the wild type (M2 mAChR WT) receptor. The higher degree of direct allosteric agonism observed at the WT by LY2033298 simply reflects higher relative expression of this construct relative to the truncated receptor.

Reviewer #2

In this manuscript, Hollingsworth et al. suggest a mechanism by which allosteric modulators for muscarinic acetylcholine receptors can achieve selectivity for the M1 subtype. The authors performed and analyzed long-scale molecular dynamics simulations of all the muscarinic receptor subtypes in different states, and hypothesize that a ‘cryptic’ binding pocket –absent in the available crystal structures– can spontaneously open in the extracellular domains of the M1 subtype to accommodate a positive allosteric modulator. According to the simulations, the opening/closing of this cryptic binding pocket in the M1 subtype is due to the labile nature of a hydrogen bond interaction between Y2.64 and C45.50. In the simulations, Y2.64 can relocate away from C45.50 (in the second extracellular loop) and towards E7.36 (in TM7) thus opening the cryptic pocket. Due to the diverse nature of position 7.36 in muscarinic acetylcholine receptor subtypes, the rearrangement of Y2.64 and opening of the cryptic binding pocket is not favored in subtypes M2-M5, thus explaining the selectivity of certain PAMs for the M1 subtype. Both the scientific hypothesis underlying this work (allosteric modulators can recognize transient binding pockets that are not necessarily visible in crystal structures) and the experimental design to test it (long scale molecular dynamics simulations and site-directed mutagenesis) are very straightforward. Also, the number and length of of the simulations performed is impressive.

We thank the reviewer for the kind words.

However, the authors use surprisingly few data from the simulations to substantiate their claims. The nature of the cryptic binding pocket is simply described in terms of the Y2.64-C45.50 distance (in Figures 2A, 3A, and 3B, which contain all the computational data in the main text, presenting data derived from this single parameter). I am convinced that there must be more information in the trajectories that can provide further insights on the nature of the cryptic pocket; it seems a colossal effort to run over 127 microseconds of simulations to monitor a single atom-atom distance.

We appreciate the reviewer’s comment, and we did indeed measure a large number of different quantities to characterize the opening of the cryptic pocket. We found one good summary statistic and, for brevity, focused on that single measurement in our original submission. In the revised manuscript, we have added a second measurement that we used to corroborate the findings of the first (tracking the hydrogen bond distance between Y2.64 and E7.36; Supplementary Figs. 3 and 12).

In addition to the two distance-based metrics for the open cryptic pocket conformation, we extensively analyzed the cryptic pocket environment. Due to the absence of a PAM in the majority of simulations, W7.34 and Y179 were free to move, which resulted in the polar core region of the allosteric site to shrink or even collapse in simulation. This significant motion made it difficult to use algorithms such as CAVER to identify the different shapes of the allosteric pocket. In addition to the distance between C45.50 and Y2.64 (Figs 2 & 3) and the distance between Y2.64 and E7.36 (Supplementary Figs. 3 and 12), we also studied the rotameric state of Y2.64 and multiple additional distances between TM2, TM7 and ECL2 to further validate the value of the presented measurements.

The site-directed mutagenesis data seem pretty straightforward, suggesting that disrupting the interaction that keeps the cryptic pocket open (with the single-point mutations Y2.64F and E7.36A) reduces the frequency of the open state, as the affinity of these mutants for the PAMs is reduced (about 1 log unit) while retaining its activity as a PAM. Despite the clarity of the goals and the presented data, I would expect a more thorough characterization of the structure/dynamics of the cryptic pocket in the MD trajectories, and perhaps a more complete analysis by site-directed mutagenesis.

We have now carried out both additional experimental work, in the form of mutagenesis experiments in the presence of an additional PAM (MIPS-1519, see new section of the manuscript entitled “*Modification of BQZ12 also confirms the importance of cryptic pocket formation for M1-selective PAM binding*” and additional panel in Figure 4), as well as new computational work in the form of simulations of the mutants analyzed in Figure 4, at the reviewer’s request. We have also added several supplementary figures including additional measurements of cryptic pocket opening (Supplementary Figure 3) and analysis of key interactions in the immediate environment (Supplementary Figs. 2 and 5).

It is conceivable that are other factors involved in the formation/stabilization of the cryptic binding pocket in the M1 muscarinic receptor.

As highlighted in our earlier study presenting the crystal structures of the inactive M1 and M4 mAChRs (Thal et al., 2016, *Nature*), there are three general thermodynamic mechanisms by which individual residues within and between orthosteric and allosteric pockets affect the interaction between such ligands—first, residues that make tighter contacts with the ligands in the closed (active) state than the open (inactive) state; second, residues that are immobilized by the binding of either ligand, such that the entropic cost is paid by the first binding event; third, non-ligand-contact residues that alter the free energy of activation of the receptor and thus the open to closed transition. However, these proposed mechanisms have been based on a limited set of solved structures and experiments analysed under equilibrium conditions. Our current study is the first to apply MD to understand any mechanism of allosteric GPCR modulator selectivity across both apo and holo receptors in both active and inactive states. Although it is indeed conceivable that additional factors related to the preceding general mechanisms may contribute to such selectivity, the fact that we observe cryptic pocket formation within the vestibular allosteric site between different states undergoing conformational transitions suggest that this is a key localized event likely representing an initial “trigger” for allosteric drug selectivity *within* a pocket, over and above any additional mechanisms that might be occurring concomitantly. Please also see below our related reply to reviewer 3 regarding cooperativity as a selectivity mechanism (point 6). We have added new text to the manuscript on page 8 to further acknowledge this possibility, but an

exhaustive investigation of all possible changes in residue dynamics throughout the receptor protein(s) is beyond the scope of the current work.

This work would also greatly benefit from an attempt (at least) to predict the existence of cryptic pockets in other receptors. If I understand properly, the authors imply that the presence of a cryptic pocket in the M1 muscarinic receptor could not have been predicted; is this so? In this case, are long-scale molecular dynamics simulations a requisite for finding such cryptic pockets? If so, their study will be restricted to just a few research groups worldwide, considerably limiting its usefulness to design subtype-specific allosteric drugs that exploit cryptic pockets. Perhaps the analysis of the MD trajectories allows a retrospective analysis on available sequences and/or crystal structures that allows to predict the presence of cryptic pockets? I understand that this may require much more computational and experimental work, but without a generalization of their results, the presented data represents a single example in a very specific protein family.

We appreciate the reviewer's comment but believe that predicting the existence of cryptic pockets in other receptors is beyond the scope of the current study. We do not feel that the utility of using MD to investigate this phenomenon will be restricted to only a few research groups. The simulations required to discover the cryptic pocket in the M1 mAChR in our study were relatively short: the pocket generally opened within 100 ns of simulation, which took a few hours on a commodity graphics processing unit that costs less than \$2000. (No supercomputer or custom computing hardware was used in our study.) Thanks to a recent decrease in cost, simulations of this length have become broadly accessible to labs worldwide, include labs that do not specialize in MD simulations (for further discussion, see Hollingsworth and Dror, 2018. "Molecular dynamics simulation for all." *Neuron*. 99:1129-43). Moreover, there are numerous approaches to applying MD to proteins used by different labs, e.g., "accelerated MD", "biased MD" etc. that can also be potentially applied to the study of cryptic pockets.

As highlighted in our manuscript, the pursuit of cryptic pockets in other protein families (e.g., kinases) has recently been acknowledged as an exciting new avenue for understanding drug selectivity. Given such prior evidence that this phenomenon has recently been recognized at other proteins, there is no reason to suspect that this cannot be operative at GPCRs — the largest and one of the most dynamic classes of drug targets — yet this has never been investigated until now. We present herein the mAChRs as the *first* such GPCR model system, because of the wealth of prior pharmacological data and available allosteric modulator tools to validate our hypothesis, and we predict that this will spur further studies in the field at different receptor systems.

In summary, the hypothesis presented in this work is novel and interesting for researchers in the area of molecular pharmacology, particularly in the field of GPCRs. While the work has the potential to be influential in these fields, the present analysis of the computational and experimental data seems too limited to fully support the claims of the authors, in particular regarding the extension of the idea of cryptic pockets for allosteric modulators in other GPCRs.

As highlighted in the preceding paragraph, we believe that our results have important implications, particularly for GPCR biology and drug development. Again, we reiterate that the role of cryptic pockets in ligand binding represents a new and very active area of research, because these pockets represent a way to identify and potentially generate drugs for otherwise “undruggable” targets, and to achieve selectivity between closely related proteins. A great deal of work has gone into discovering such pockets, both experimentally and computationally. However, no one has shown the one can achieve selectivity at a GPCR—by far the largest class of drug targets—by exploiting a cryptic pocket. This is of tremendous interest for the drug discovery field in general, because one-third of all drugs target GPCRs, and a major problem with GPCR-targeted drugs is that they often bind off-target receptors that are closely related to their target. As also highlighted above, the methods presented here are generalizable to many other proteins, though the application of those methods in those systems will require further experimental and computational work. Our goal in this study is to present the method of discovery and underlying mechanism of one such cryptic pocket at an allosteric site in order to help inspire further work in the field. We have revised the title, introduction and conclusion to better reflect our intention.

Reviewer #3

The manuscript „Cryptic pocket formation underlies subtype selectivity of GPCR allosteric modulators“ by Ron Dror and Arthur Christopoulos and colleagues reports on the identification of a ‘cryptic pocket’ in the allosteric binding site of the muscarinic M1 receptor (M1R). This pocket is not visible in the inactive crystal structure of the M1R but forms spontaneously as assessed by microsecond-timescale computational simulations. Interestingly, this pocket seems to form far more often in the M1R than in other muscarinic receptor subtypes. The authors claim that this ‘cryptic pocket’ dictates subtype selectivity of a previously characterized M1R positive allosteric modulator (PAM), i.e. BQZ12, a structural analogue of the well-characterized M1R PAM BQCA. Based on these data the authors aim to generalize their findings and suggest that subtype selectivity of allosteric modulators can be achieved by exploiting differences in the conformational ensemble of GPCRs.

Overall, this is a very interesting manuscript. Novel strategies to design subtype selective drug candidates are needed and this manuscript offers a powerful approach by combining cutting-edge computational methods with more classical pharmacology. From a methodological point of view, the approach undertaken here will likely have a great impact on other GPCRs and related drug targets.

We are grateful for the reviewer’s kind words on our study.

While I find the approach very intriguing, powerful and of broad interest, I am more skeptical about some specific conclusions drawn from the data and about the generalizability of specific findings of this manuscript.

1) I am not convinced that cryptic pocket formation is independent of the state (apo, inactive inverse agonist-bound, active agonist-bound) of the M1R. My concern is based on two lines of evidence:

a. GPCR activation involves a process called ‘allosteric coupling’ where conformational changes at the intracellular part of the receptor (e.g. outward tilt of TM6) are allosterically linked to the conformation of the ligand binding pocket and extracellular domains (PMID: 27362234). In particular, the M2R undergoes major extracellular

conformational changes which result in a complete closure of the ligand binding pocket including a ‘contraction’ of the allosteric binding site (also visible in Figure 1b vs 1c of the current manuscript). However, the authors state that the ‘geometry of the allosteric site in an active-state model ... is similar’ (page 7, lines 123-124) to the one seen in the inactive crystal structure. This is in contrast to all available active structures which show a (more (e.g. M2R) or less (e.g. β 2AR) pronounced) contraction of the ligand binding pocket and rearrangement of extracellular domains. How ‘active’ is the MIR active-state model reported here? Please describe, show and validate the active state MIR model. This would be an interesting finding and should be highlighted.

As the reviewer points out, the allosteric site does indeed contract when the muscarinic receptors transition from their inactive-state conformation to their active-state conformation. However, the allosteric pocket maintains its overall planar shape during this transition. Until the cryptic pocket opens, it is not possible to dock BQZ12 to the allosteric site in a way that agrees with experimental data in either the inactive or active states.

We were also surprised that the overall conformational state of the receptor appears to have no effect on the dynamics of cryptic pocket formation within the vicinity of the allosteric modulator, despite the conformational transitions experienced by the remainder of the protein. However, we carefully studied the simulations of M1 we carried out not only in different conformational states but also in the presence and absence of different orthosteric ligands as well as under different protonation states in the active state. Our analysis indeed reveals that the dynamics of the cryptic pocket are unaffected by the larger conformational changes that take place upon activation.

As described in our Methods section, the active state models of M1, M3 and M4 mAChRs were generated through homology modeling to the experimental active state M2 mAChR structure for the transmembrane core while using the loops from the respective inactive state structures. To ensure that each active state receptor remained in an active conformation (a conformation that could couple to a G protein), we first overlaid each of the active state models with the available β 2-Gs GPCR-G protein complex. We then placed restraints on all residues that would be within 5 angstroms of the G protein to keep the receptor active for the full length of the simulation. Through this combination of modeling and simulation restraints, we can ensure that the active states remain in a conformation that can couple to a G protein throughout the simulation.

We have included a new supplementary figure in order to highlight the differences between the active and inactive state models (Supplementary Fig. 13).

We believe that the ability of our active-state models to accurately predict the outcomes of mutagenesis studies provides significant validation for our modeling work. If our models were not accurate, we would not expect them to have much predictive power. To further validate the predictive power of our models, we ran additional active-state simulations of the M1 mAChR mutants studied in Figure 4. The resulting simulations show little to no opening of the cryptic pocket, consistent with the reduction in binding affinity observed experimentally, further validating the predictive power of our active-state models.

b. BQZ12 cannot be docked into the inactive MIR crystal structure (e.g. page 5; lines 86-88). However, the authors have successfully docked BQZ12 into an active state MIR model in a previous paper (PMID: 25326383). According to this, one would expect structural

changes in the allosteric binding site to occur upon receptor activation (see above). This is contradictory to the authors' statement that the 'geometry of the allosteric site in an active-state model ... is similar' (page 7, lines 123-124) to the one seen in the inactive crystal structure. Please clarify.

Some of the authors of this manuscript docked BQZ12 into an active state model of the M1 mAChR in a previous paper, but the pose for BQZ12 that was proposed at that time does not agree with the experimental mutagenesis data presented in the paper itself (an unfortunate situation that all of the authors acknowledge). In particular, the previously predicted pose did not make close contact with Y2.64, which mutagenesis showed to play a key role in binding. In the absence of an open cryptic pocket, it is not possible to dock BQZ12 into either the active-state or inactive-state models in a way that agrees with the experimental data. Once the cryptic pocket opens, however, BQZ12 can be docked into both the active-state or inactive-state models in a way that agrees with the experimental data.

While the allosteric site does undergo a contraction upon activation, our work here shows that the dynamics of the cryptic pocket itself are unaffected by these larger changes (see our response to the previous comment for more detail). We have revised the manuscript to clarify this observation.

2) On page 13, lines 261-263 the authors state 'We predicted that each of these mutations would reduce the frequency of cryptic pocket opening and thus reduce the affinity of the non-planar, M1-selective PAMs such as BQZ12 and BQCA'. Whereas the pharmacological data show indeed that mutation of Y2.64F and E7.36A reduce PAM affinity, the manuscript does not provide evidence that the frequency of cryptic pocket opening is reduced by the mutations. Additional molecular dynamics simulations of both receptor mutants bound to ACh and BQZ12 should be run to provide data for the above-mentioned statement.

We agree that this would make an excellent addition to the study. We have carried out additional simulations of the three experimentally tested mutants in the Active, ACh-bound and Active, apo conditions.

In each of the three studied M1 mAChR mutants (E7.36A, E7.36S, and Y2.64F), the disruption of the cryptic pocket stabilizing hydrogen bond between Y2.64 and E7.36 disfavors the open cryptic pocket conformation in the allosteric site. As our computational work would predict that BQZ12 or BQCA binding would require the open cryptic pocket, these results (which are also in agreement with the experimental mutagenesis) further support our proposed mechanism of selectivity.

3) Figure 4: Many of the presented results have been published earlier. This is not uncommon as the same mutation can be studied under totally different perspectives in different manuscripts. However, in the case of BQCA, there are published results which contradict the findings presented here: in a previous report by the same group (PMID: 25326383), the influence of the Y2.64F mutation on BQCA affinity has been studied. It was concluded that this mutation does not influence BQCA affinity ($pK_b(BQCA)$: 4.82 ± 0.06 and 4.66 ± 0.00 at wildtype and Y2.64F mutant receptors, respectively). In the current manuscript the authors show that this mutation does indeed significantly reduce BQCA affinity (Figure 4b. $pK_b(BQCA)$: 5.23 ± 0.05 and 4.76 ± 0.23 at wildtype and Y2.64F mutant receptors, respectively). It seems that the affinity of BQCA to the free receptor

varies across studies, e.g. 4.82 ± 0.06 (PMID: 25326383), 4.49 ± 0.09 (PMID: 24443568) and 5.23 ± 0.05 (this study). Please clarify these contradictory results.

It is now well acknowledged that experimental conditions can have significant effects on the pharmacological properties of ligands, both orthosteric and allosteric. It is not uncommon for the affinities, potencies, efficacies and cooperativity values to fluctuate depending on the assay buffer, the time on incubation and the temperature the assays were run at. In PMID: 25326383, the radioligand binding assays were performed at 37°C for 1.5 hours, in a supplemented binding buffer, using [³H]QNB as the radiolabelled orthosteric antagonist. In these conditions, the affinity of BQCA for the M1 receptor allosteric site on the unoccupied receptor was indeed quantified at 4.49 ± 0.09 . In PMID: 24443568, the radioligand binding assays were performed at +4°C for 4 hours, in an identical binding buffer as PMID: 25326383, using [³H]NMS as a radiolabelled orthosteric antagonist. In these conditions, the affinity of BQCA for the free receptor was 4.92 ± 0.09 . In the current manuscript, we used a more simplified binding buffer, lacking glucose, potassium and bicarbonate salts. Additionally, our binding assays were performed at room temperature for 16 hours to ensure that equilibrium is reached between orthosteric and allosteric ligands. Consequently, our estimated affinity of BQCA for the allosteric site of the M1 mAChR is slightly altered compared to previously described, $pK_B = 5.23 \pm 0.05$. Nonetheless, such modest differences between different experimental conditions for a low affinity interaction are to be expected and are still marginal, with the variance of the affinity estimates from all the aforementioned studies being a little over 5-fold. Essentially, all these studies are in general accord for an affinity of BQCA at the *unoccupied* M1 mAChR allosteric site of approximately 0.6 – 3 μ M.

4) On page 8, line 147-149 the authors state that ‘simulations of BQZ12 in this pose remained stable in the allosteric site in multiple independent microsecond-timescale simulation’. Where are these data? I just find a list of simulations times in Supplementary Table 3. It would be very important to show these simulations and describe and assess the frequency and stability of interactions between BQZ12 and receptor residues from the cryptic pocket and residues found to interact with BQZ12 in a previous publication (PMID: 2532638).

We agree that these simulations merit further highlighting in the manuscript. We have included a new supplementary figure (Supplementary Fig. 7) which further highlights the dynamics and stability of BQZ12 in the open-cryptic-pocket conformation of the allosteric site.

5) The data provided in this manuscript do not convince me that cryptic pocket formation is a general mechanism which applies to many PAMs at the MIR, related muscarinic subtypes, or even other Class A GPCRs. The authors mention several times that BQZ12 ‘and related/similar PAMs (e.g. page 4, line 73; page 5, line 89; page 5, line 94) are nonplanar in nature (page 11, 220-221)’. However, the data shown in the manuscript are solely based on BQZ12 itself and BQCA (although there are contradictory results in the literature for BQCA, c.f. point 3). Which ‘related’ PAMs do the authors refer to? It would be very important to demonstrate both computationally and pharmacologically that these ‘related’ PAMs bind to the cryptic pocket. Otherwise, the findings reported here are a special case for BQZ12 (and maybe BQCA) at the MIR.

To date, all known M1-selective allosteric modulators share the nonplanar shape of BQZ12, as opposed to the planar shape of LY2119620 that non-M1-selective PAMs share. These similar M1-specific molecules include BQZ12 and BQCA (Abdul-Ridha et al., *J Biol Chem* 289:33701–11, 2014), as well as recently disclosed compounds from Merck (Beshore et al., *ACS Med Chem Lett* 9:652–6, 2018), Vanderbilt (Conn et al., *Trends Pharmacol Sci* 30:148–55, 2009), Pfizer (Davoren et al., *J Med Chem* 59:6313–28, 2016), Takeda (Sako et al., *Neuropsychopharmacology*, 2018) and Monash (Mistry et al., *J Med Chem* 56:5151–72, 2016). We have now added references to all of these compounds on pages 5 and 12 of the revised manuscript.

Furthermore, to experimentally demonstrate that the findings from BQZ12 apply to the other similarly shaped PAMs, we have now included additional pharmacological data for MIPS-1519 (highlighted in Figure 4 and in a new experimental section of the manuscript), which represents a structure that contains a “midway” extension from BQCA to BQZ12 in the predicted non-planar arm.

6) *The authors state in the title that ‘cryptic pocket formation underlies subtype selectivity’. However, subtype selectivity of BQCA has been shown to arise from positive cooperativity with ACh at MIR rather than high affinity. Please specify what mechanism of selectivity underlies which PAM.*

“Affinity” and “cooperativity” are actually thermodynamically related to one another in an allosteric mechanism (Christopoulos, 2002, *Nature Rev. Drug. Discover.*). As defined in the Methods section, the cooperativity, commonly designated as “ α ”, is actually the *ratio* of the dissociation constant of a modulator for the free receptor (i.e., $K_{B(Apo)}$) to that of the agonist-occupied receptor (i.e., $K_{B(ACh)}$). We have chosen to present our findings in terms of unconditional dissociation constants ($K_{B(Apo)}$ and $K_{B(ACh)}$), but the difference between the two (i.e., $pK_{B(ACh)} - pK_{B(Apo)}$) equals $\log \alpha$, i.e., the logarithm of the cooperativity. Thus, if $K_{B(ACh)}$ is 1 nM, but $K_{B(Apo)}$ for the free allosteric site is 1 μ M, then the cooperativity (α) is 1000 ($\log \alpha = 3$).

When we refer to “affinity” as a “mechanism for selectivity,” we are specifically referring to significantly different values of K_B for the *free* allosteric site between receptors, such that a modulator binds to one receptor subtype, even in the absence of orthosteric agonist, while not binding appreciably to others; this has been the traditional view for many years. When we refer to “cooperativity” as a mechanism of selectivity, we are specifically referring to a modulator binding with *similar* affinities to relatively conserved allosteric sites between receptor subtypes (i.e., similar $K_{B(Apo)}$ values), but with markedly *different* affinities *in the presence of agonist* (i.e., different $K_{B(ACh)}$ values). The mechanisms underlying these two different scenarios remain to be determined but represent a logical extension for future studies of GPCR allostery.

7) *Figure 2: I suggest to add a second, more mechanistic, surrogate parameter to describe the dynamics of ‘cryptic pocket formation’. Estimation of the distance between E7.36 and Y2.64, i.e. the formation of the H-bond depicted in Figure 2C, would be helpful in this regard.*

We agree that the inclusion of this surrogate parameter is helpful to understand more mechanistically what is happening at the cryptic pocket. We have chosen to include such a figure as a new supplementary figure (Supplementary Fig. 2). We have also used this new measurement

as a secondary metric for the opening of the cryptic pocket in two additional figures (Supplementary Figs. 3 and 12).

We chose not to include this secondary measurement in Figure 2 as we feel that the inclusion of both distances in the Figure 2A trace panel would disrupt the easy-to-follow nature of the figure. We have, however, included a pointer to the requested trace in the Figure 2 caption.

8) Figure 1: The authors state that the allosteric pockets 'are similar in shape' based on the shape of the pocket surface. How is this pocket surface measured exactly? To me the pockets do look fairly distinct, e.g. the pocket surfaces of M2R, M3R and M4R, but not M1R, extend between TM5 and TM6; and the pocket surface of M4R covers part of the space between TM2 and TM7. On what parameters is the degree of similarity based? Another report claimed that the allosteric pockets are distinct (PMID: 27490498). Please discuss.

Our description of pocket similarity was originally based on visualization of the pocket shape and sizes in the inactive-state structures. It was originally expected that these allosteric sites would vary greatly between subtypes. While there are small differences between the inactive-state structures, the degree to which they are similar has been striking.

The study cited by the reviewer was published when experimental structures were available only for the M2 and M3 mAChRs. The authors thus relied on homology modeling and simulation to model the other mAChRs. In addition, the timescale of the simulations (~10 ns) was likely insufficient to allow the models to equilibrate fully following the extensive modeling required to build the homology models. Our present study used more recently solved inactive-state structures for the M1 and M4 mAChRs, used the inactive-state structure to better inform our active-state models of the M1, M3 and M4 mAChRs, and carried out simulations 100-fold longer than those reported previously.

9) The authors state that 'allosteric sites are often very similar structurally across receptor subtype' (page 3, lines 52-53). However, this statement can only be drawn from muscarinic receptor structures. The wording suggests that this would be a general feature of all GPCRs. Please rephrase.

Actually, as highlighted in our recent review, Thal et al. (*Nature* 559:45–53, 2018), structural biology studies are now confirming that the locations of allosteric sites can be observed to be conserved not only within closely related GPCRs, such as the mAChRs, but also (surprisingly) between less conserved families. A striking example of the latter, for instance, is an intracellular allosteric site that has been observed not only at CCR2 and CCR9 chemokine GPCRs, but also at a similar spot in the β_2 adrenergic receptor (Thal et al., *Nature*, 2018). Given this surprising finding, the question then becomes how such structurally conserved regions between disparate GPCRs show high selectivity. One possible mechanism is the dynamic formation of cryptic pockets as described herein. This review is cited in the Introduction (ref. 8), and we have modified the sentence accordingly.

10) The title suggests that cryptic pocket formation would be a general mechanism for all GPCR allosteric modulators. However, the data provided in this manuscript do not support

this statement (see above). Based on the data, cryptic pocket formation would be, at best, a mechanism applicable to BQZ12 and BQCA. Please rephrase.

Although we believe that what we are presenting is the first proof-of-concept example of a likely more widespread phenomenon, we agree with the reviewer for the need to be more circumspect and have thus retitled the manuscript accordingly.

Reviewers' comments:

Reviewer #1 (Remarks to the Author):

I appreciate the improvements and additional data included in this version. My points 3 and 4 have been solved/answered. However, with the additional data provided I feel that I have more concerns regarding the atomistic mechanism by which the cryptic pocket forms. So I'm still not convinced on my points 1 and 2.

1) Although the authors claim in their response that "the interaction E7.36-K1.27 breaks spontaneously" in all inactive-state simulations, the newly added SFig 5 shows that this is not the case. In fact, in 4 out of 6 simulations in the inactive state this interaction exists far more often than it breaks. This then worries me of whether it is OK not having K1.27 in the active M1 model.

2) Despite the changes in Figure 2, I still feel that showing one selected single replica in Figure 2 does not give the complete picture of the opening of the cryptic site. I'm worried that a reader looking only the figures that are now in the main text would get the impression that the mechanism by which the cryptic pocket forms is simpler than it is. The fact that the distance distributions for 2.64-7.36 in the new Suppl Fig 3 are not bimodal suggests that these residues are involved in other interactions as well. On the search for such additional interactions I found that it is not clear from the crystal structures that Y2.64 interacts with the backbone carbonyl of C45.50, as additional candidates are possible including the backbone NH of C45.50, the side-chain of E/N45.49 or aromatic interactions with the Trp in ECL1. So I'm wondering if the interaction C45.50-Y2.64 alone is a good indicator of a closed pocket. Definitely, additional characterization in terms of other distances, side-chain dihedrals of Y2.64 and 7.36, and helix shifts/rotations would provide a much better idea of the mechanism.

Reviewer #2 (Remarks to the Author):

In this revised manuscript, the authors include a more thorough analysis of the molecular dynamics simulations and provide more computational and experimental data. Together with the changes in the text, I consider that the authors have addressed satisfactorily the issues that I raised in my first review, and I recommend the publication of this manuscript in Nature Communications.

Reviewer #3 (Remarks to the Author):

The authors have done a fine job in addressing all of my comments. I believe the manuscript has substantially improved in both experimental evidence and clarity in the presentation of the results. I have no further comments and I strongly recommend publication of this manuscript.

Response to Reviewers' Comments:

We thank all of the reviewers for their thoughtful feedback on our resubmission and appreciate that the majority of reviewers recommend publication of the manuscript in its previously submitted form. Below, we have included the reviewers' comments (indented and in italics), followed by our responses.

Reviewer #1

I appreciate the improvements and additional data included in this version. My points 3 and 4 have been solved/answered.

We are happy that we were able to address these concerns.

However, with the additional data provided I feel that I have more concerns regarding the atomistic mechanism by which the cryptic pocket forms. So I'm still not convinced on my points 1 and 2.

1) Although the authors claim in their response that "the interaction E7.36-K1.27 breaks spontaneously" in all inactive-state simulations, the newly added SFig 5 shows that this is not the case. In fact, in 4 out of 6 simulations in the inactive state this interaction exists far more often that it breaks. This then worries me of whether it is OK not having K1.27 in the active M1 model.

The interaction between E7.36 and K1.27 does break spontaneously (at least once) in all of our inactive-state simulations. We did not claim that it is usually broken in all inactive-state simulations; our point was simply that it breaks frequently, suggesting that it is not a particularly strong interaction.

More importantly, this salt bridge—even when formed—does not prevent the cryptic pocket from opening. In simulations of the inactive state, E7.36 can interact with both K1.27 and the rotated Y2.64 in the presence of the open cryptic pocket. In other words, breakage of this salt bridge is not required for pocket opening. The fact that we see the same pocket opening frequency in both active-state and inactive-state simulations (without and with K1.27, respectively) also supports this point.

At the reviewer's request, we added K1.27 into our active-state model. This resulted in two key observations. First, the positions of TM1 and TM7 in the M1 active-state model (based on the experimentally solved M2 active-state structure) are such that K1.27 is too far from E7.36 to make a direct interaction (5.8 Å from the charged nitrogen in K1.27 to the closest oxygen of E7.36, compared to the 2.98 Å distance observed in the inactive-state M1 mAChR structure). Second, when we started simulations of active-state M1 with K1.27 and E7.36 positioned as close to one another as possible, the K1.27–E7.36 interaction occasionally formed, but it was broken most of the time, as shown in the plot below. The cryptic pocket opened frequently in these simulations, even during periods when the K1.27–E7.36 interaction was formed (e.g., the final part of the light-green simulation in the plot below).

We note in the revised manuscript (p. 9) that “The crystal structure of the M1 mAChR shows a salt bridge between E7.36 and K1.27, but this salt bridge breaks frequently in simulation and does not prevent rotation of E7.36 (Supplementary Fig. 5).”

2) Despite the changes in Figure 2, I still feel that showing one selected single replica in Figure 2 does not give the complete picture of the opening of the cryptic site. I’m worried that a reader looking only the figures that are now in the main text would get the impression that the mechanism by which the cryptic pocket forms is simpler than it is. The fact that the distance distributions for 2.64-7.36 in the new Suppl Fig 3 are not bimodal suggests that these residues are involved in other interactions as well. On the search for such additional interactions I found that it is not clear from the crystal structures that Y2.64 interacts with the backbone carbonyl of C45.50, as additional candidates are possible including the backbone NH of C45.50, the side-chain of E/N45.49 or aromatic interactions with the Trp in ECL1. So I’m wondering if the interaction C45.50-Y2.64 alone is a good indicator of a closed pocket. Definitely, additional characterization in terms of other distances, side-chain dihedrals of Y2.64 and 7.36, and helix shifts/rotations would provide a much better idea of the mechanism.

At the reviewer’s request, we now show two simulations (replicas) in Fig. 2 of the revised manuscript.

As discussed in our previous response to reviewers, we have extensively characterized the interactions around the cryptic pocket to identify any and all interactions that play a significant role in determining cryptic pocket opening. Indeed, our analysis does show that the mechanism is relatively simple.

The additional interactions that the reviewer points out do not play significant roles in the opening of the cryptic pocket beyond the descriptions already present in the text. The hydrogen bond between Y2.64 and C45.50 does fluctuate between the backbone carbonyl and amide of C45.50 and the hydroxyl group of Y2.64. However, it is the presence of a hydrogen bond between Y2.64 and C45.50, not the particular backbone atom of C45.50 that is participating in the interaction, that keeps the pocket closed to the allosteric site. Indeed, much of our analysis has been carried out on the smallest distance between Y2.64 and C45.50 in order to account for the fluctuating nature of this interaction. Residue 45.49, which is also near the pocket, was also not observed to play a

significant role in keeping the pocket open or closed in the vast majority of simulations. In a single M4 simulation replicate, an interaction was formed between 45.49 and Y2.64, following a slight conformational change, that resulted in the pocket remaining closed; this is already described in our manuscript. Finally, we have not been able to identify a role for W23.50 in pocket opening and believe that it does not contribute to the pocket opening mechanism.

As stated in our previous response, we had previously studied the torsion angle of Y2.64 to probe pocket opening. While the results tracked with the pocket opening, the noisy nature of the metric led us to favor the Y2.64–C45.50 distance metric to track pocket opening. We have also used various metrics to investigate helix rotation and movement, but the results proved noisy and ambiguous.

We believe that the distance metric between Y2.64 and C45.50 does indeed capture the true mechanism of pocket opening at the M1 mAChR. This is supported not only by all of the simulation results across M1–M4 mAChRs but also by the accompanying experimental mutagenesis data.

Nevertheless, we have included the following caveat in the revised manuscript (p. 9): “While it is possible that additional factors, such as relative helix positioning or the conformation of ECL2, play a role in the formation of the cryptic pocket, the motions of Y2.64 and E7.36 appear essential in governing the stabilization of the open pocket conformation.”

Reviewer #2

In this revised manuscript, the authors include a more thorough analysis of the molecular dynamics simulations and provide more computational and experimental data. Together with the changes in the text, I consider that the authors have addressed satisfactorily the issues that I raised in my first review, and I recommend the publication of this manuscript in Nature Communications.

Reviewer #3

The authors have done a fine job in addressing all of my comments. I believe the manuscript has substantially improved in both experimental evidence and clarity in the presentation of the results. I have no further comments and I strongly recommend publication of this manuscript.

We appreciate the kind words from both reviewers 2 and 3 and are glad to have addressed all of their questions and concerns.